# Relationship between Attachment to Pet and Post-Traumatic Growth after Pet Loss: Mediated Moderating Effect of Cognitive Emotion Regulation Strategy through Separation Pain

**DOI:** 10.3390/bs12080291

**Published:** 2022-08-18

**Authors:** Hyo Jin Park, Goo-Churl Jeong

**Affiliations:** 1Institute of Well-Being Health Psychology, Sahmyook University, Seoul 01795, Korea; 2Department of Counseling Psychology, College of Health and Welfare, Sahmyook University, Seoul 01795, Korea

**Keywords:** pet loss, attachment, separation pain, post-traumatic growth, cognitive emotion regulation strategy

## Abstract

The declining fertility rate and an aging population have accelerated the number of single-person households and nuclear families, and the number of households raising pets has naturally increased. However, pet owners experience great sorrow and trauma due to the death of their pets. The stronger the attachment to pets, the more severe the separation pain caused by pet loss. The purpose of this study was to analyze the moderating effect of a cognitive emotion regulation strategy mediated through separation pain on the relationship between attachment and post-traumatic growth after pet loss among owners. The study participants were 303 owners who have experienced pet loss. We analyzed the mediated moderating effects by PROCESS macro. The results showed that the adaptive cognitive emotion regulation strategy strengthened the effect of attachment to pets on post-traumatic growth and decreased the effect on separation pain. Conversely, the maladaptive cognitive emotion regulation strategy weakened the effect of attachment to pets on post-traumatic growth and strengthened the effect on separation pain. The act of intentionally expanding the perspective on pet loss experience, switching into a more positive focus, and accepting reality will reduce the grief of its companions and become an opportunity for growth.

## 1. Introduction

In our society, low fertility rate and an aging population continue to intensify, resulting in an increase in single-person households and nuclear families. An increase in single-person households naturally leads to an increase in households living with pets [1]. In addition, it has been reported that the adoption rate of pets has increased due to changes in daily life and lockdown policies around the world caused by the COVID-19 pandemic [2]. The social distancing policy has increased owners’ relationship with pets. According to previous studies, 67% of American households have pets in the United States, and the pet industry in China has grown by a whopping 2000% over the past 10 years [3]. According to statistics from the Ministry of Agriculture, Food, and Rural Affairs [4], the number of households raising companion animals in Korea also increased by 1.81 million compared to 2015, with about 6.38 million households raising companion animals. This increase in pet owners is not simply a result of the development of a new industry but is closely related to a change in the attitude toward pets.

Along with the increase in pet owners, the perspective on pets is also changing. Moving away from the concept of a “pet” that is taken and raised as an animal, it has established itself as a “companion,” emphasizing the meaning of a companion living together [5]. Beyond raising pets out of simple curiosity, owners accept them as members of the family, share many aspects of their lives, and feel happy through their relationships with pets [6]. The positive effects of raising pets have been demonstrated in several previous studies. Well [7] reported that raising pets improves physical health and helps one communicate with others. In other words, pet caregivers feel less lonely than those without pets, and their interpersonal relationships become more active [8]. It has also been reported that elderly people living with pets experience less depression than those who do not [9]. Through these results, we found that relationships with pets play an important role in raising positive emotions. In addition, since pets provide unconditional affection to humans, people can relieve tension and stress through interactions with pets and increase happiness by obtaining emotional stability [10,11]. Now, pets are not simply being raised but are considered beings who sympathize with and pursue happiness. 

However, despite the myriad of benefits that pets bring, their death can bring unavoidable pain. Compared to humans, pets have a relatively short lifespan; therefore, humans inevitably experience pet loss during the rearing process [12]. It is very painful to lose someone who has been with you for a long time, regardless of the reason. A person who has experienced such loss experiences difficulties in daily life, including emotional and physical difficulties [13]. Of course, sadness and mourning after the loss of a loved one is a normal and natural reaction; however, it may cause pathological problems if it interferes with daily life or if one does not recover even after a long time. Although there are individual differences in the degree and extent of maladaptive psychological problems that individuals experience due to loss, they typically experience traumatic pain, such as anger and guilt, and separation pain, such as longing, for about six months after the loss, and then they return to daily life. It is common to return, but depending on the person, the state of sadness and pain continues to be maintained, and it remains pathological and causes difficulties in daily life [14]. This difference is also greatly influenced by the relationship between pets and coping styles with grief [15]. 

Recently, many reports have shown that not all people who have experienced pain or trauma undergo posttraumatic stress [16,17]. A person can experience positive changes in the process of overcoming the pain caused by trauma, called post-traumatic growth. Although the probability of being exposed to trauma in our lifetime is close to 80%, the probability of experiencing post-traumatic stress disorder (PTSD) is only approximately 5% [18]. This shows the inner strength of human beings to protect or recover from trauma, suggesting that trauma can be used as a scaffolding for further growth [19]. According to previous studies, firefighters and first responders who encounter numerous traumatic events due to the nature of their jobs also showed positive changes through the rumination process [20]. In addition, bereaved families who have lost their loved ones due to large-scale disasters, such as the 9/11 attack, have overcome the loss through religious significance [16]. Parents who lost their children also experienced post-traumatic growth after a terrible experience, such as discovering a new meaning of life by reconstructing the meaning of death [17].

Such post-traumatic growth can have a greater effect as more active and adaptive cognitive coping is performed [21]. The deep sadness and separation pain felt after the loss of a pet makes it impossible to rationally reflect on current events. The cognitive emotion regulation strategy is to go beyond the simple repetitive recall of the grief caused by the loss of a pet and to coordinate it at a higher mental level. Post-traumatic growth does not occur automatically and can be promoted through intentional and active intervention. This method of intentionally controlling emotions through cognitive intervention is called a cognitive emotion regulation strategy. A cognitive emotion regulation strategy, which refers to the cognitive handling of emotionally awakened information, is expected to play an important role in regulating emotions caused by pet loss and linking it with post-traumatic growth [22]. Cognitive emotion regulation strategies accept events as they are, expand their perspectives, and lay the groundwork for growth through positive reevaluation. Rather than a simple and repetitive rumination on the traumatic event, the cognitive process of analyzing, meaning, and reevaluating traumatic events promotes post-traumatic growth [23].

In general, the stronger the love for and attachment to pets, the greater the pain caused by pet loss. According to a previous study on attachment to pets, people with a high level of anxiety attachment to pets had a lower level of acceptance of their death and a higher level of death-related pain and self-blame for their death [24]. This suggests that if a person experiences the death of a pet, one needs to take deliberate cognitive measures, such as accepting it and not blaming oneself or others. Thus, although traumatic experiences caused by pet loss cause separation pain, the use of cognitive emotion regulation strategies can help alleviate psychological maladjustment [25]. In fact, as a result of conducting an online survey of rearers in Australia and the UK who lost their pets, it was found that the higher the cognitive emotion regulation strategy, the fewer symptoms of sadness, resentment, trauma, and guilt caused by the pet loss [26]. As a result of a survey of victims who survived the great flood, the more catastrophic and frequent the rumination, the higher the PTSD symptom score; the victims who used cognitive emotion regulation strategies, such as positive re-evaluation, experienced post-traumatic growth [27].

Attachment, which is a strong bond with pets, entails great separation pain after pet loss and may affect post-traumatic growth. However, separation pain may be reduced and promote post-traumatic growth improved through acceptance of death, broadening of perspectives, and positive re-evaluation using adaptive cognitive emotion regulation strategies. Therefore, in this study, we tried to verify the moderating effect of cognitive emotion regulation strategies on the relationship between attachment to pets and post-traumatic growth after pet loss among owners, and to verify whether it also moderates the mediating effect of separation pain.

Research Purpose and Research Model

The purpose of this study was to examine the moderating effect of a cognitive emotion regulation strategy mediated through separation pain in the relationship between attachment to pets and post-traumatic growth after experiencing pet loss among owners. The research model is illustrated in Figure 1.

## 2. Materials and Methods

### 2.1. Participants

The participants of this study were 303 Korean adults who experienced pet loss. Among the study participants, 133 (43.9%) were male, and 170 (56.1%) were female. The average age of the participants was 34.7 years (*SD* = 12.9) and ranged from 19 to 69 years old. There were 203 (67.0%) non-religious participants and 100 (33.0%) religious participants. In terms of marital status, 205 (67.7%) were single, and 98 (32.3%) were married. The most common lost pets were dogs at 216 (71.3%), cats at 38 (12.5%), and other animals at 49 (16.2%). Disease was the most common cause of pet loss (123; 40.6%), followed by natural death (106; 35.0%), accidental death (54; 17.8%), and loss of pets (20; 6.6%). After pet loss, 90 participants (29.7%) no longer had a pet, but 175 people (57.8%) got the same kind of pet again, and 52 people (17.2%) had the same pet, even by breed. After pet loss, 178 people (58.7%) perceived that they had received sufficient support from their families.

### 2.2. Procedure

This study was approved by the Institutional Review Board of Sahmyook University (IRB No. SYU 2022-01-012-001). The survey was conducted between 8 March and 5 April 2022 through an online survey in a non-face-to-face manner. The questionnaire was created in Google Forms, and a description of the study participants was posted on the first page of the questionnaire, including information on the purpose of the questionnaire and how to participate. In the explanatory text, basic guidance on the research, measures to protect personally identifiable information, and the scope of use of the collected data were described. After reading the explanation, it was clarified that there was no penalty for not participating in the survey or not completing it. The online survey was designed so that only those who read all explanations and agreed to voluntarily participate in the survey could start it. After the survey, an online gift certificate was provided as a reward for participation time.

### 2.3. Instrument

#### 2.3.1. Attachment to Pet

The Korean version [28] of the Lexington Attachment to Pets Scale (LAPS) by Johnson, Garrity, and Stallones [29] was used. The LAPS consists of 23 questions, and responses are made on a 4-point Likert scale. The higher the total score on the scale, the stronger the attachment to the companion animal. In this study, the Cronbach’s α of the scale was found to be as high as 0.94.

#### 2.3.2. Separation Pain

The Korean version [30] of the separation pain factor in the Inventory of Complicated Grief-revised (ICG-r) scale by Prigerson and Jacobs [31] was used. It consists of 18 questions, and responses are recorded on a 5-point Likert scale. A higher score indicates a higher level of pain experienced by the loss of companion animals. In this study, the Cronbach’s α value of the scale was found to be as high as 0.98.

#### 2.3.3. Cognitive Emotion Regulation Strategy

The Korean version [32] of the Cognitive Emotion Regulation Questionnaire by Garnefski, Kraaija, and Spinhoven [23] was used. It consists of 36 questions, answered on a 5-point Likert scale. There are a total of nine sub-factors, which are divided into adaptive strategies (expanding perspective, rethinking plans, positive focus change, acceptance, and positive re-evaluation) and maladaptive strategies (self-blame, criticism of others, rumination, and catastrophe). The higher the score, the stronger the use of each cognitive emotion regulation strategy. In this study, the Cronbach’s α value of the adaptive strategy was 0.89, and that of the maladaptive strategy was 0.94.

#### 2.3.4. Post-Traumatic Growth

The Korean version [33] of the post-traumatic growth scale by Tedeschi and Calhoun [34] was used. It consists of 16 questions, answered on a 5-point Likert scale. The sub-factors consist of the following: change in self-perception, increase in interpersonal depth, discovery of new possibilities, and increase in spiritual interest. The higher the score, the higher the post-traumatic growth caused by pet loss. In this study, the Cronbach’s α of the scale was found to be as high as 0.92.

### 2.4. Data Analysis

The collected data were analyzed using IBM SPSS Statistics for Windows (version 25.0; IBM Corp., Armonk, NY, USA). Descriptive statistics of the variables were calculated, and the Cronbach’s α coefficient was used to assess the reliability of the scale. Differences in study variables according to general characteristics were analyzed by an ANOVA, and the post hoc test was analyzed using Scheffé’s method. The mediated moderating effect of cognitive emotion regulation strategies through separation pain on the relationship between attachment and post-traumatic growth was analyzed using PROCESS Macro Release 3.5 (Model 8) [35].

## 3. Results

### 3.1. Descriptive Statistics of Research Variables

We calculated the mean, standard deviation, skewness, and kurtosis values of attachment to the pet, cognitive emotion regulation strategy, separation pain after pet loss, and post-traumatic growth (Table 1). The absolute values of skewness and kurtosis of the study variables were all less than 1.06, assuming normality of the research variables.

### 3.2. Difference between Separation Pain and Post-Traumatic Growth after Pet Loss According to General Characteristics

Table 2 shows the differences in post-traumatic separation pain and post-traumatic growth according to general characteristics using the ANOVA. Regarding sex, males had significantly higher separation pain than females (*F* = 33.61, *p* < 0.001), and females had significantly higher post-traumatic growth than males (*F* = 6.50, *p* = 0.011). The religious group had a significantly higher level of separation pain (*F* = 31.30, *p* < 0.001), but there was no significant difference in post-traumatic growth. In terms of the type of pet, the group that lost cats had significantly higher separation pain (*F* = 17.43, *p* < 0.001), and the group that lost dogs had significantly higher post-traumatic growth (*F* = 5.85, *p* = 0.003). There were no statistically significant differences on pet loss type. Separation pain was significantly higher in the group that raised the same type of pet of the same breed after pet loss than in the other groups (*F* = 6.23, *p* < 0.001). In contrast, the group that felt they received sufficient support from their family after pet loss had significantly lower separation pain (*F* = 55.94, *p* < 0.001) and significantly higher post-traumatic growth than the group that did not (*F* = 12.13, *p* = 0.001).

### 3.3. Mediated Moderating Effect of the Cognitive Emotion Regulation Strategy through Separation Pain

To analyze the mediating moderating effect of the cognitive emotion regulation strategy, first, its moderating effect on the relationship between attachment to pets and post-traumatic growth was analyzed. Then, its moderating effect on the relationship between attachment to pet and separation pain among mediating pathways was analyzed. The result of the analysis of the moderating effect of adaptive strategy among the cognitive emotion regulation strategies is presented in Table 3, and the results of the analysis of the moderating effect of a maladaptive strategy are presented in Table 4. Before analyzing the moderating effect, we controlled for gender and age among the general characteristics of the study participants in all regression models.

Model 1 in Table 3 shows that the moderating effect of the adaptive cognitive emotion regulation strategy on the relationship between pet attachment and post-traumatic growth was statistically significant (*B* = 0.39, *p* = 0.009). The more adaptive the cognitive emotion regulation strategy used, the stronger the effect of attachment to pets on post-traumatic growth after pet loss. The moderating effect of the adaptive cognitive emotion regulation strategy is shown in Figure 2a. However, when used together with the adaptive cognitive emotion regulation strategy, separation pain had a significant positive effect on post-traumatic growth (*B* = 0.13, *p* = 0.009).

Model 2 in Table 3 shows that the adaptive cognitive emotion regulation strategy (*B* = −0.78, *p* < 0.001) had a significant moderating effect on the relationship between pet attachment and separation pain. The higher the adaptive cognitive emotion regulation strategy, the weaker the effect of attachment to pets on separation pain after pet loss. To help understand the moderating effect, the moderating effect of the adaptive cognitive emotion regulation strategy is shown in Figure 2b. The result of the moderated mediating effect analysis is presented in Figure 2c with path coefficients. The index of moderated mediation of the indirect effect of attachment to pets on post-traumatic growth through separation pain was significant at the 95% confidence level (*Index* = −0.10, 95% *CI* [−0.22, −0.02]). The adaptive cognitive emotion regulation strategy was found to significantly control for the indirect effect of attachment to pets.

Model 1 in Table 4 shows that the moderating effect of the maladaptive cognitive emotion regulation strategy on the relationship between attachment to pets and post-traumatic growth was statistically significant (*B* = −0.60, *p* < 0.001). The more the maladaptive cognitive emotion regulation strategy was used, the weaker the effect of attachment to pets on post-traumatic growth after pet loss. The moderating effect of the maladaptive cognitive emotion regulation strategy is shown in Figure 3a.

Model 2 in Table 4 shows that the maladaptive cognitive emotion regulation strategy had a significant moderating effect on the relationship between pet attachment and separation pain (*B* = 0.27, *p* = 0.002). The higher the maladaptive cognitive emotion regulation strategy, the stronger the effect of attachment to pets on separation pain after pet loss. The moderating effect of the maladaptive cognitive emotion regulation strategy is shown in Figure 3b. The result of the moderated mediating effect analysis is presented in Figure 3c with path coefficients. When used together with the maladaptive cognitive emotion regulation strategy, separation pain had a significant negative effect on post-traumatic growth (*B* = −0.18, *p* = 0.036).

The index of moderated mediation of the indirect effect of attachment to pets on post-traumatic growth through separation pain was not significant at the 95% confidence level (*Index* = −0.05, 95% *CI* [−0.12, 0.01]). The maladaptive cognitive emotion regulation strategy did not significantly control for the indirect effect of attachment to pets.

## 4. Discussion

This study examined a cognitive emotion regulation strategy that can protect the mental health of caregivers from the loss of pets and pursue growth through the experience of pet loss.

The stronger their attachment to their pet, the more pet owners experience intense sadness and pain after pet loss, but they can also grow through the process of overcoming this pain. In this study, it was found that the stronger the attachment to the pet, the greater the growth after its loss. This process confirmed that the adaptive cognitive emotion regulation strategy strengthened the effect of attachment to pets on post-traumatic growth. It has been reported that the adaptive cognitive emotion regulation strategy increases life satisfaction by affecting it in the direction of giving or promoting meaning to life [36]. Cognitive strategies for companion animal caregivers to accept pet loss as they are, broaden their perspectives, change their focus in a positive direction, and reevaluate the current situation can boost post-traumatic growth. This process was similarly demonstrated in the process of mourning for people. Based on previous studies on mourning, if we reconstruct the meaning of death after experiencing the loss of a loved one or family member, we can overcome the shock of loss and gain the strength to live a new life by adapting to the present [37].

This study found that the stronger the attachment to pets, the higher the separation pain felt after pet loss. If a person experiences too much separation pain due to the death of one’s companion animal, one may not be able to adjust to daily life, and this may lead to a threat to one’s mental health, such as depression. In general, 20% of people who have lost a family member need professional help [38]. This is because the traumatic experience of losing a family member is so powerful and complex that it poses a threat to the physical and mental health of those left behind, and it can increase the risk of developing depression or disease [39]. If a person has a strong attachment to a pet, they experience strong separation pain, similar to losing their family, which may hinder new beginnings and growth. Due to preoccupation with the thoughts and memories of the lost companion animal, it is difficult to concentrate on other tasks, and there is a feeling that something important in life is missing or that a part of oneself is lost [40]. Therefore, it is necessary to help prevent the experience of losing a companion animal from interfering with an individual’s daily life and growth through active intervention.

The results of previous studies that cognitive emotion regulation strategies can effectively control negative emotions, such as emotional pain caused by loss, partially support the results of this study [41]. Adequate separation pain due to pet loss, along with adaptive cognitive emotion regulation strategies, increase post-traumatic growth. However, considering the results of this study that maladaptive cognitive emotion regulation strategies aggravate separation pain and that separation pain had a negative effect on post-traumatic growth, it was confirmed that excessive separation pain was a factor inhibiting post-traumatic growth. The maladaptive cognitive emotion regulation strategy consists of rumination, self and other criticisms, and catastrophic factors. The act of repeatedly reflecting on death, blaming oneself and others, and the catastrophic thought that everything is over deepened the pain of separation and strongly hindered post-traumatic growth. Therefore, although separation pain caused by pet loss cannot be completely eliminated, post-traumatic growth can be expected if the positive cognitive and emotional regulation strategy is adjusted so that the person does not experience excessive separation pain. We can promote specific positive cognitive emotion regulation strategies through professional counseling. In other words, the attitude of acceptance that accepts the current situation as it is the starting point for solving all problems. When we experience painful events in our lives, we can train ourselves to develop an attitude to accept them without avoiding them. Furthermore, we can reduce the pain by shifting our focus to other tasks or enjoyable things instead of what happened to us. Even though we have lost a pet, we need to improve our skills to find and reevaluate positive factors, such as making ourselves recall how happy our pets were to meet us. Although the death of a pet is the saddest thing at this point in time, it actually trains us to expand our view that more difficult things can happen in our lives. Furthermore, we can check to specifically plan and implement life after the loss of a pet again. We can have the same kind of pet again, or we can try raising a different kind of pet. In addition, there was a case in which subjective well-being was improved through the method of observing and communicating with pets online [42]. Therefore, you do not necessarily have to adopt a new pet. In this way, post-traumatic growth can be promoted through the promotion of positive cognitive emotion regulation strategies applicable to reality.

The death of a companion animal is accompanied by great suffering, similar to the death of a family member, but efforts to cognitively accept these events, expand one’s perspective, and focus on a positive direction help companion animal caregivers grow. As a result of the qualitative analysis of the post-traumatic growth of people who experienced pet loss, recognition of the appreciation of life, personal strength, and new possibilities were high [43]. These results support the notion that positive cognitive–emotional regulation strategies support post-traumatic growth. This study is meaningful in that it revealed that the stronger the attachment to the pet, the stronger the experience of separation pain caused by pet loss, but the intervention of a positive cognitive emotion regulation strategy increased post-traumatic growth despite separation pain.

The limitations of this study and suggestions for follow-up studies are as follows. Due to the COVID-19 pandemic, data were collected mainly from online club gatherings that raise pets. Therefore, no survey was conducted among the elderly who were not accustomed to using computers or smartphones. In the case of the elderly, they do not live alone voluntarily, but live alone because their children have grown up and become independent. Their experience of identifying the meaning of and relationship with pets may be different than that of adults. In a follow-up study, we propose to conduct a study on the experience of pet loss among the elderly. In addition, since the subject of the study was the experience of pet loss, participants may have experienced pain in the process of recalling their past experiences. Therefore, careful consideration is required when obtaining information regarding the experience of loss and separation pain, and it is important to conduct a survey while respecting the participant’s situation and emotions. However, in the case of online surveys, there was a limit to paying direct attention to and being more sensitive to the research participant’s feelings. In follow-up studies, it is necessary to provide individual help so that research participants can participate in the study in a comfortable state.

Despite the above limitations, this study is meaningful in that it presents a method for healthy acceptance, growth, and maturity when obstacles in daily life occur through pet loss experience. Based on the results of this study, it is expected that it will be possible to provide practical help by preparing a pet loss pre-education program centered on cognitive emotion regulation strategies for pet caregivers.

## Figures and Tables

**Figure 1 behavsci-12-00291-f001:**
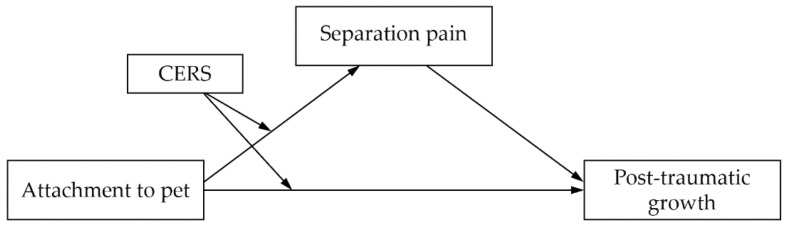
Research model. CERS = Cognitive Emotion Regulation Strategy.

**Figure 2 behavsci-12-00291-f002:**
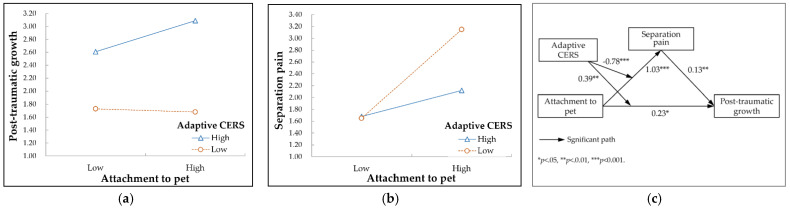
Moderating effect of adaptive cognitive emotion regulation strategy (CERS). (**a**) Moderating effect of adaptive CERS on the relationship between attachment to pet and post-traumatic growth after pet loss; (**b**) Moderating effect of adaptive CERS on the relationship between attachment to pet and separation pain after pet loss; (**c**) Research model with path coefficients.

**Figure 3 behavsci-12-00291-f003:**
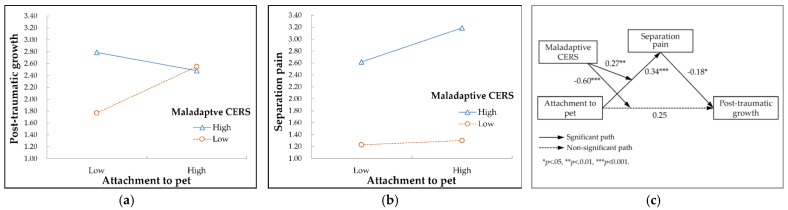
Moderating effect of a maladaptive cognitive emotion regulation strategy (CERS). (**a**) Moderating effect of a maladaptive CERS on the relationship between attachment to pet and post-traumatic growth after pet loss; (**b**) Moderating effect of a maladaptive CERS on the relationship between attachment to pet and separation pain after pet loss; (**c**) Research model with path coefficients.

**Table 1 behavsci-12-00291-t001:** Descriptive statistics of research variables (*n* = 303).

Variables	*M* ± *SD*	Skewness	Kurtosis
Attachment to pet	3.39 ± 0.47	−0.89	0.15
Separation pain	2.13 ± 1.17	1.06	−0.04
Cognitive Emotion Regulation Strategy			
Adaptive CERS	2.75 ± 0.72	0.37	−0.18
Maladaptive CERS	2.50 ± 0.97	0.43	−0.58
Post-traumatic Growth	2.29 ± 0.97	0.86	0.29

CERS = Cognitive Emotion Regulation Strategy.

**Table 2 behavsci-12-00291-t002:** Analysis of the difference between separation pain and post-traumatic growth after pet loss according to general characteristics (*n* = 303).

Variables	Categories	Separation Pain		Post-Traumatic Growth	
		*M* ± *SD*	*F*(*p*)	*M* ± *SD*	*F*(*p*)
Gender	Male	2.55 ± 1.43	33.61 (<0.001)	2.13 ± 0.94	6.50 (0.011)
	Female	1.81 ± 0.78		2.41 ± 0.98	
Religion	None	1.88 ± 0.91	31.30 (<0.001)	2.35 ± 0.98	2.18 (0.141)
	Have	2.65 ± 1.46		2.17 ± 0.94	
Species of pet	Dog ^a^	1.93 ± 0.93	17.43 (<0.001)	2.39 ± 0.97	5.85 (0.003)
	Cat ^b^	3.07 ± 1.58	a,c < b	1.84 ± 0.73	
	Etc. ^c^	2.29 ± 1.41		2.17 ± 1.05	
Causes of pet loss	Disease	2.19 ± 1.20	0.58 (0.630)	2.35 ± 0.96	0.45 (0.719)
	Natural	2.03 ± 1.12		2.25 ± 0.99	
	Accident	2.25 ± 1.26		2.20 ± 0.89	
	Missing	2.03 ± 1.11		2.40 ± 1.14	
Pet after pet loss	None ^a^	1.99 ± 1.12	6.23 (<0.001)	2.13 ± 0.96	2.23 (0.085)
	Same species & breed ^b^	2.68 ± 1.37	a,c,d < b	2.27 ± 1.02	
	Same species ^c^	2.14 ± 1.15		2.31 ± 0.92	
	Different species ^d^	1.69 ± 0.77		2.61 ± 1.03	
Family support	Insufficient	2.69 ± 1.38	55.94 (<0.001)	2.06 ± 0.88	12.13 (0.001)
	Sufficient	1.75 ± 0.80		2.45 ± 1.00	

^a,b,c,d^ Alphabet is the result of a post hoc test using Scheffé’s method.

**Table 3 behavsci-12-00291-t003:** Moderating effect of an adaptive strategy through separation pain on the relationship between attachment to pet and post-traumatic growth after pet loss (*n* = 303).

Model	Predictor	Separation Pain			Post-Traumatic Growth			*R* ^2^
		*B*	*SE*	*p*	*B*	*SE*	*p*	
1	Attachment to pet				0.23	0.11	0.036	0.414
	Separation pain				0.13	0.05	0.009	
	Adaptive CERS				0.80	0.07	<0.001	
	Interaction *				0.39	0.15	0.009	
2	Attachment to pet	1.03	0.11	<0.001				0.462
	Adaptive CERS	−0.34	0.07	<0.001				
	Interaction *	−0.78	0.17	<0.001				

* The interaction term was composed of the product of attachment to the pet and the adaptive cognitive emotion regulation strategy. CERS = Cognitive Emotion Regulation Strategy.

**Table 4 behavsci-12-00291-t004:** Mediated moderating effect of a maladaptive strategy through separation pain on the relationship between attachment to pet and post-traumatic growth after pet loss (*n* = 303).

Model	Predictor	Separation Pain			Post-Traumatic Growth			*R* ^2^
		*B*	*SE*	*p*	*B*	*SE*	*p*	
1	Attachment to pet				0.25	0.13	0.060	0.168
	Separation pain				−0.18	0.09	0.036	
	Maladaptive CERS				0.24	0.10	0.013	
	Interaction *				−0.60	0.13	<0.001	
2	Attachment to pet	0.34	0.09	<0.001				0.740
	Maladaptive CERS	0.84	0.04	<0.001				
	Interaction *	0.27	0.09	0.003				

* The interaction term was composed of the product of attachment to the pet and the maladaptive cognitive emotion regulation strategy. CERS = Cognitive Emotion Regulation Strategy.

## Data Availability

The data presented in this study are not available elsewhere.

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
