# Peer review of "Relationship between Attachment to Pet and Post-Traumatic Growth after Pet Loss: Mediated Moderating Effect of Cognitive Emotion Regulation Strategy through Separation Pain"

_behavsci, 2022, doi:10.3390/bs12080291_

Round 1

Reviewer 1 Report

Title: Relationship Between Attachment to Pet and Post-traumatic 2 Growth After Pet Loss: Mediated Moderating Effect of Cognitive Emotion Regulation Strategy Through Separation Pain

Introduction

The following sentence about post-traumatic stress needs elaboration. “Recently, many reports have shown that not all people who have experienced pain or trauma undergo posttraumatic stress.”

The authors talk about trauma and pain following the loss of a pet. The literature clearly does support this.  The authors seem to suggest that this is post-traumatic stress.

Please clarify this.

Methods

“2.3.4 Post-traumatic growth 175

Growth after pet loss was measured using a post-traumatic growth scale [34,35]. It 176 consists of 16 questions answered on a 5-point Likert scale.”

It is suggested that the authors include the authors of the scale here.

Discussion

4. Discussion 292

“During the COVID-19 pandemic, the number of people raising pets has increased. 293 Currently, pets are not considered merely as animals but as family members that communicate with people. However, inevitable pet loss causes psychological pain in caregivers.”

It is suggested that the authors delete these sentences in the discussion.  They are not necessary

‘The stronger their attachment to their pet, the more pet owners experience intense sadness and pain after pet loss, but they also grow through the process of overcoming this.” 

Consider rewording

The stronger their attachment to their pet, the more pet owners experience intense sadness and pain after pet loss, but they can also grow through the process of overcoming this pain.

I believe this sentence is too strong and should be reworded:

If these negative emotions are not resolved and persist for a long time, they may develop post-traumatic stress disorder, which may lead to suicidal thoughts and death [41].

The authors state the following:

“The results of previous studies that cognitive emotion regulation strategies can effectively control negative emotions, such as emotional pain caused by loss, partially support 330 the results of this study [43]. Adequate separation pain due to pet loss, along with adaptive cognitive emotion regulation strategies, increase post-traumatic growth. Therefore, although separation pain caused by pet loss cannot be completely eliminated, post-traumatic growth can be expected if the cognitive and emotional regulation strategy is adjusted so that the person does not experience excessive separation pain.”

This is very important – it is the main point of the article.  It would improve the manuscript if the authors could talk about, in more detail, what cognitive emotional regulation strategies are.  Can these skills be taught?  If so, how?

Reviewer 2 Report

The topic of the article is very valuable, and the authors have done a great job on the research question. However, there are still the following problems that need to be modified. Frist,the authors need to explain in detail what relevant studies have been done on post-traumatic stress of pet owners after pet loss, and why the authors choose CERs as the regulatory variable, so as to further elaborate the theoretical contribution of the article. Second, the authors need to continue to strengthen the elaboration of the applicability of the article in the discussion part, and discuss how the owner can better get out of the trauma after pet losing. Third, please supplement the following reference.

     Zhou, Z.; Yin, D*.; Gao, Q*. Sense of Presence and Subjective Well-Being in Online Pet Watching: The Moderation Role of Loneliness and Perceived Stress. International Journal of Environmental Research and Public Health, 2020, 17, 9093. 
